# Assessing the Influence of Food Insecurity and Retail Environments as a Proxy for Structural Racism on the COVID-19 Pandemic in an Urban Setting

**DOI:** 10.3390/nu14102130

**Published:** 2022-05-20

**Authors:** Rachael D. Dombrowski, Alex B. Hill, Bree Bode, Kathryn A. G. Knoff, Hadis Dastgerdizad, Noel Kulik, James Mallare, Kibibi Blount-Dorn, Winona Bynum

**Affiliations:** 1Center for Health and Community Impact, Division of Kinesiology, Health & Sport Studies, College of Education, Wayne State University, Detroit, MI 48202, USA; bodebr@wayne.edu (B.B.); kathryn.gray@wayne.edu (K.A.G.K.); gm4209@wayne.edu (H.D.); ab7564@wayne.edu (N.K.); jpmallare@wayne.edu (J.M.); 2Center for Urban Studies, College of Liberal Arts and Science, Urban Health, Wayne State University, Detroit, MI 48202, USA; er2818@wayne.edu; 3Detroit Food Policy Council, Detroit, MI 48226, USA; kibibi@detroitfoodpc.org (K.B.-D.); winona@detroitfoodpc.org (W.B.)

**Keywords:** structural determinants, healthy retail, food access, food security, social determinants, social policy, nutrition, built environment

## Abstract

A collaborative partnership launched the Great Grocer Project (GGP) in March 2021 in Detroit, Michigan where health inequities, including deaths due to COVID-19, have historically been politically determined and informed by socially entrenched norms. Institutional and structural racism has contributed to a lack of diversity in store ownership among Detroit grocers and limited access to high-quality, affordable healthy foods as well as disparate food insecurity among Detroit residents. The GGP seeks to promote Detroit’s healthy grocers to improve community health and economic vitality through research, programs, and policies that have the potential to advance health equity. A cross-sectional design was used to explore relationships between scores from the Nutrition Environment Measures Surveys-Stores (NEMS-S) in 62 stores and city-level data of COVID-19 cases and deaths as well as calls to 211 for food assistance. Regression and predictive analyses were conducted at the ZIP code level throughout the city to determine a relationship between the community food environment and food insecurity on COVID-19 cases and deaths. COVID-19 cases and deaths contributed to greater food insecurity. The use of ZIP code data and the small sample size were limitations within this study. Causation could not be determined in this study; therefore, further analyses should explore the potential effects of individual grocery stores on COVID-related outcomes since a cluster of high-scoring NEMS-S stores and calls to 211 for food security resources inferred a potential protective factor. Poor nutrition has been shown to be associated with increased hospitalizations and deaths due to COVID-19. It is important to understand if a limited food environment can also have a negative effect on COVID-19 rates and deaths. Lessons learned from Detroit could have implications for other communities in using food environment improvements to prevent an uptick in food insecurity and deaths due to COVID-19 and other coronaviruses.

## 1. Introduction

In global public health practice, protective factors are sought to mediate and moderate health outcomes, especially among populations of people that have been historically underserved [1]. For example, having access to high-quality healthy foods is a protective factor for developing obesity [2]. To understand and unpack protective factors for healthy eating behaviors, it is important to craft public health interventions with an equity lens and to also consider the social and structural components of racism that influence health outcomes, such as COVID-19 and nutrition [3]. 

In the United States (U.S.), there have been nearly 80 million recorded cases of COVID-19 and approximately 950,000 COVID-related deaths, with 1/3 of all COVID-related deaths associated with obesity (obesity is framed by the historical clinical representation of, majority, white bodies; therefore, association is a more relevant term which helps to account for unjust framing of the built environment where segregation and structural racism are known barriers in access to land, parks, and healthful, affordable nutritious options) [4,5,6,7]. Globally, 39% of people are overweight and of those, 13% are obese [7]. In the U.S., 30.7% of people are overweight, 42.4% are obese, and 9.2% are severely obese [8,9]. One in three adults (2.37 million people) throughout the world experience food insecurity, or a lack of access to nutritious food, and food insecurity is most often concentrated within low-income, racial and ethnic minority communities, which has also been exacerbated by the COVID-19 pandemic [10,11], (p. 190).

COVID-19, food insecurity, and obesity have followed similar inequitable trends among historically underserved populations in U.S. communities [12,13,14]. The number of COVID-19 cases reported in the U.S. and sorted by race have been higher among those that identify as American Indian/Alaskan Native, Black/African American, and Hispanic/Latino than those that identify as non-Hispanic White or Asian/Pacific Islander [5]. COVID-19 death rates reported and sorted by race have impacted American Indian/Alaskan Native, Black/African American and Hispanic/Latino populations two times more than those who identify as non-Hispanic White [5,14]. In the U.S., in 2019, adults that identified as Black/African American (22.5%) or Hispanic/Latino (18.5%) reported food insecurity at nearly double the mean rate (10.5%) of the collective U.S. adult population [15,16]. The estimated prevalence of food insecurity impacting people in the U.S., given the pandemic, was highest among people that identified as Black/African American (21%) and significantly less among people that identified as White (11%) [16,17]. The prevalence of obesity and severe obesity is highest among U.S. adults that identify as Black/African American (49.6%), Hispanic/Latino (44.8%), and non-Hispanic White (42.2%) [13]. Furthermore, when nutrition-related health outcomes and COVID-19 cases in the U.S. are explored together by race/ethnicity those who identified with an ethnic minority group were reported to have more severe cases of COVID-19 and were three times as likely to be hospitalized when nutritional-related comorbidities (i.e., obesity, diabetes) were also present [5,6,18,19]. Similarly, individuals in poverty have been shown to have greater risk for COVID-19 infection, hospitalizations and death [20,21].

In Michigan (MI), COVID-19 rates, hospitalizations and deaths followed similar trends as the rest of the nation. The death rates attributed to COVID-19 and sorted by race impacted African American/Black and American Indian/Alaskan Native Michiganders more often than White Michiganders [5,22]. At the onset of COVID-19, people that identified as Black/African American were experiencing more deaths associated with COVID-19 than any other racial/ethnic group [5]. Obesity rates among Michiganders also mirror national trends, with Hispanic/Latino (46.7%) and African American/Black (42.7%) MI residents being diagnosed with obesity at higher rates than White MI residents (34.2%) [23,24]. Nearly two million MI residents, a majority of whom identify as African American/Black and Native American, experience food insecurity [24]. Wayne County, MI, which includes the city of Detroit, has had the most COVID deaths in the state [22]. Wayne County also has the second highest COVID rates in the state, second only to Oakland County (a border county to Detroit) [22,25,26]. Within Wayne County, residents who identify as African American/Black experienced the greatest number of COVID-related cases (53.8%) and deaths (83.7%), leaving many communities disparately impacted [25,26]. The overall food insecurity rate in Wayne County prior to the pandemic was over 30% and this rate increased post-pandemic [25]. Obesity prevalence among Detroit adults is 38% and disproportionately impacts Black/African American Detroit residents [27,28]. 

### Socially and Structurally Determined Outcomes: COVID-19, Food Insecurity, and Obesity

Social determinants of health (SDOH) and structural racism work in tandem as a predictor for negative health-related quality-of-life outcomes [3,29,30]. Historical components of structural racism (e.g., redlining, blue-lining, U.S. land grabs, inequitable hiring practices, access to health care, and segregation) have informed modern elements of structural racism (e.g., employment type, cultivating wealth, health insurance, and culturally relevant health communications) when considering ZIP-code-level effects [3,31,32,33,34]. Elements of structural racism have been traced to COVID-19 case rates, hospitalizations, death rates, and lack of or slow distribution of COVID-19 vaccinations [35,36]. This syndemic catalyzed inquiries of structural racism on health, and outlined linkages between structural racism and food insecurity, where food insecurity and inequitable food environments could be considered a proxy for structural racism on health outcomes such as obesity and COVID-19 cases and deaths [15,32,37,38,39,40,41,42].

The ties of structural racism are closely linked to the historical and modern barriers of food insecurity, such as blue-lining, land grabs, and nutrition policy implementation, which were strategic structural efforts to keep specific populations from fully participating in health-promoting activities (e.g., supermarket ownership and access to supermarkets, land ownership/farmers, and nutrition program participants) [15,31,32,35,43]. One structurally problematic act of racism is blue-lining, which trickles into the capacity for specific populations to acquire, maintain, and sustain healthy retail environments, which are vital to improving nutrition-related community health outcomes [31,44]. Blue-lining has kept healthy retail options geographically out of reach from specific population groups, namely those whose primary language was not English and/or those that were not White [30,44].

Healthy eating habits and retail environments are dynamically related since individual behavior, society, and structural determinants influence the uptake of nutrition options [45,46,47]. People who live near an abundance of convenience stores compared to healthy grocery stores have a significantly higher prevalence of obesity [47]. This is especially true for impoverished families, who often have limited food access and purchasing options and are burdened by many diet-related diseases as well as food insecurity [48,49]. Therefore, creating and supporting healthy food environments specifically within disadvantaged health areas is essential for improving healthy eating behaviors [50,51]. The community food environment, in particular food retail environments, contributes to community health [52,53] and represents a promising venue to improve healthy eating behaviors [53,54,55]. Furthermore, having high-quality, affordable healthy food available in food retail settings allows people to make healthier choices under the premise that structural and social factors have contributed to a more equitable presence of healthier foods and can play a substantive role in preventing obesity and the conditions related to food insecurity. [39,50,53,54,55,56]. 

The city of Detroit is a health-disparate city and has over 60 full-service grocery stores [57]. While large chain grocery stores have returned to more affluent areas of the city, most residents continue to rely on neighborhood independently-owned grocers for food purchases [58]. Often within independent grocery stores, the variety and quality of food and beverages for sale falls considerably below that of chain grocers [57,58]. Detroit has subsequently seen an increase in the number of dollar stores accompanied by a net decrease in independent, full-line grocery stores [57,58,59,60]. While there are plans for the implementation of two Black-owned retail environments in Detroit, at present, there are no Black-owned grocery stores in the city, which outlines a snapshot of the disadvantage experienced by Black/African American business owners in securing loans for small business ventures [57,59,60]. To address these issues, the Detroit Grocery Coalition launched the Great Grocer Project (GGP) in March 2021 to advance health equity by improving the grocery environment within the city, as well as healthy eating behaviors among Detroit residents. A component of this project is to assess the food environment within grocery stores using the Nutrition Environment Measures Survey (NEMS). The NEMS scores in combination with customer surveys, store participation in community events and state-level food safety ratings were combined to create a five-tier score for each of the GGP stores. The GGP has been promoting the highest-scoring stores over the past year to increase healthy food sales within these stores among Detroit residents. 

There are limited reports in the literature regarding the connection between improving community food environments as a response to moderating structural racism and understanding proxy measures of structural racism, such as food insecurity, and its potential impact on mitigating the rate of COVID-19 incidence and deaths. The large population of African Americans, Latino/Hispanics, and immigrant families within Detroit, and the high prevalence of obesity and food insecurity [25,28,61] renders Detroit as a purposeful setting in which to conduct this research. Additionally, this research attempts to utilize food insecurity and unhealthy food environments as a proxy measure for structural racism given the inequitable differences among Black/African American residents of Detroit who experience food and nutrition insecurity as a result of racist and discriminatory historical acts and policies [57,62,63]. Furthermore, this study provides a first assessment of the protective factors healthy grocery environments can undertake in reducing the number of COVID-19 cases and deaths as well as improving food security among racial and ethnic minority communities within urban settings. 

## 2. Materials and Methods

A cross-sectional approach was used to explore relationships between scores from the Nutrition Environment Measures Surveys (NEMS) [64,65] in 62 grocery stores, city-level data of COVID-19 cases and deaths, and calls to 211 for food assistance. NEMS scores were calculated using modified methods from the original NEMS scoring sheet [65]. Three sub-scores were calculated to obtain the total NEMS score: quality, availability, and price. Since five new fruit items and four new vegetable items were added to the adapted NEMS survey, these were included in the calculations of quality and availability. Additionally, analysis of lean meat availability changed from including only sirloin ground beef to including sirloin, ground turkey and other lean meats (<10% fat). Quality scores evaluate the percentage of acceptable quality of the fresh produce available in stores. Availability was assessed by indicating which, and how many, healthy options were available in each store. Price scores were assigned by giving positive scores to healthier items that were available at a cheaper price than the less-healthy items. If the healthier option was more expensive than the alternative, the item was scored with a negative value. These scores were summed in their respective categories to obtain quality, availability, and price scores along with the total sum for the NEMS score.

Cumulative case counts, cumulative death counts and both death and case rates by ZIP code were obtained from the City of Detroit’s open data portal with a database on COVID-19 from 1 February 2020 to April 2022 [26]. Publicly available data on food assistance were obtained from the United Way for Southeastern Michigan website [66]. The number of calls to 2-1-1 for food assistance since 1 March 2020, were acquired. 

ZIP codes in low-income, urban areas comprising primarily Black/African American Detroit residents [67] were used to examine the tracts that may be most impacted by structural racism. Therefore, mean scores by ZIP code were obtained for NEMS scoring and for use in the analyses. Additionally, COVID-19 cases and death counts and rates were included to examine the health disparities associated with structural racism among the identified ZIP codes. The case and death rates were defined as cases or deaths per 100,000 residents. The Wayne State University Institutional Review Board (IRB) approved this study (IRB: 065117B3X). 

## 3. Analysis

*Regressions.* The calls to 211 for food assistance, publicly available COVID-19 case rates, death rates, cumulative death and case counts, and mean NEMS scores within each ZIP code were included in the analyses. Multiple regression analyses were conducted at the ZIP-code level (N = 27) throughout the city to determine a relationship between the community food environment on COVID-19 cases and deaths. For the first analysis, calls to 211 for food assistance (“food insecurity”) were used as the dependent variable and COVID-19 cases, deaths, and average NEMS store scores (average store scores per ZIP code) were compared as independent variables under the assumption that COVID-19 has caused greater food insecurity and local grocery stores were a potential protective factor. The second regression used COVID-19 death rates as the dependent variable and store scores, food insecurity (calls to 211), and case rates as predictors. All assumptions were tested prior to completing regression analyses. Regressions were selected to determine the relationships between food insecurity, COVID-19, and NEMS scoring to ultimately observe structural racism in this geographical area [68].

*Spatial Analysis.* Local Indicators of Spatial Autocorrelation (LISA) provide information related to the location of spatial clusters and outliers and the types of spatial correlation. Local statistics are important because the magnitude of the spatial autocorrelation was not necessarily uniform over the study area [69,70]. Significance was tested by comparison to a reference distribution obtained by random permutations [69]. This analysis used 999 permutations to determine the differences among the spatial units. A positive value for the local Moran’s I index indicates that a feature has neighboring features that have similarly high or low attribute values, meaning that it is a part of a cluster. A negative value indicates that a feature has neighboring features that have dissimilar values, indicating that this feature is an outlier. 

In either circumstance, the *p* value for the feature must be small enough for the cluster or outlier to be considered statistically significant. LISA enables distinctions to be made among a statistically significant (0.05 level) cluster of high values (HH), a cluster of low values (LL), an outlier in which a high value is surrounded mostly by low values (HL), and an outlier in which a low value is surrounded mostly by high values (LH). In addition, for the value of a z-score larger than +1.96, the outcomes are defined as clusters with both (HH) and (LL). If the value of a z-score is less than −1.96, the outliers are considered clusters with (HL) and (LH). The global and local Moran’s I statistics with Empirical Bayesian rates were calculated using GeoDa 1.18.0 10 December 2020 [71], an open-source spatial analysis system.

## 4. Results

In Wayne County, there were 18,302 calls for food assistance, and 12,173 (66.5%) of those calls came from within the City of Detroit and the ZIP codes representing this sample. The mean number of calls for food assistance in the sample was 585.76 (SD = 271.68). Additionally, the case results of the NEMS scoring and ranges for the full scale and subscales are displayed in Table 1. The mean score of NEMS for this sample was 25.53 (SD = 6.21) (Table 1). Further, COVID-19 death and case counts were significantly correlated (*R* = 0.78, *p* < 0.001), but death and case rates were not (*R* = 0.11, *p* = 0.32).

### 4.1. Regressions

#### 4.1.1. Regression 1: Predicting Food Insecurity

The assumptions for the first regression were met. Partial regression plots demonstrated linear relationships between the dependent variable and independent variables. Visual inspection of scatterplots demonstrated homoscedasticity. VIF scores were less than 10 and tolerance scores were greater than 0.1, demonstrating no multicollinearity. Studentized deleted residuals verified that there were no outliers to the data. The Durbin–Watson value of 1.79 supported independence of residuals of observations. Based on leverage points and Cook’s Distance, there were no highly influential data points. The distribution was normal as indicated with the P-P plot of standardized residuals. 

The model predicting food insecurity demonstrated good model fit (*R*^2^ = 0.78; Adjusted *R*^2^ = 0.74). The overall model was statistically significant; case count, death count, and NEMS scores significantly predicted calls to 211 for food assistance (*F*(3,17) = 20.13, *p* < 0.001). Significance was demonstrated with cumulative case count and showed a positive relationship between COVID-19 case counts and calls to 211. Additionally, COVID-19-related deaths significantly predicted calls for food assistance. Based on ZIP code, as case counts of COVID-19 increased, so did calls for food assistance, indicating higher levels of food insecurity. Though the slope coefficient was not significant, higher NEMS scores were negatively related to food insecurity, which may suggest that grocery stores with higher quality, price, and availability scores are a protective factor against food insecurity in the City of Detroit. Further analyses are needed to explore this potential effect and, therefore, NEMS scores were kept in the final regression model (Table 2). The regression equation was as follows: Predicted Food Insecurity (Y) = (136.97) − (0.356 NEMS) + (0.047 Case Counts) + (1.89 Death Counts).

#### 4.1.2. Regression 2: Predicting Death Rates from COVID-19

Partial regression plots demonstrated linear relationships between the dependent variable and independent variables. Visual inspection of scatterplots demonstrated homoscedasticity. VIF scores were less than 10 and tolerance scores were greater than 0.1, demonstrating no multicollinearity. Studentized deleted residuals verified that there were no outliers to the data. The Durbin–Watson value of 1.38 supported independence of residuals. Based on leverage points and Cook’s Distance, there were no highly influential data points. The distribution was normal as indicated with the P-P plot of standardized residuals. There was no evidence of multicollinearity in the second analysis as calls to 211, case rates, and NEMS scores were not significantly correlated. The second regression did not demonstrate significance or good model fit (*R*^2^ = 0.19, adjusted *R*^2^ = 0.06, *F*(3,17) = 1.40, *p* = 0.28), indicating that death rates are not well predicted by case rates, NEMS scores, or food insecurity (Table 2).

#### 4.1.3. Spatial Analysis: Local Clustering with Empirical Bayes Rates

The Moran’s I test for autocorrelation returned significant results for death rates and NEMS scores (pseudo *p*-value = 0.018) with similar results for case rates (pseudo *p*-value = 0.019). Death rates and calls to 211 for food assistance were also significant (pseudo *p*-value = 0.002). Higher NEMS scores and higher numbers of calls to 211 were significantly clustered (pseudo *p*-value = 0.001) with the highest Moran’s I at 0.582 out of all the models ran. 

The LISA spatial analysis highlighted the varied impact of COVID-19 and food insecurity within the City of Detroit. Different areas of the city experienced different levels of disease burden, with the most high and low clusters appearing in northwestern and central Detroit where there is a higher population and higher income (in central Detroit) compared to Detroit overall. The LISA results with high and low clusters showed that population density was not simply a confounding factor in the existence of clusters (Figure 1, Figure 2 and Figure 3). The LISA analysis also showed that food assistance calls to 211 could serve as a rapid diagnostic tool during times of crises or when more granular data are not available.

## 5. Discussion

The results revealed a significant relationship between food insecurity and COVID-19 cases and deaths. As these relationships were positive, increases in COVID-19 cases and deaths led to higher levels of food insecurity. This was also true in the spatial analyses with more clusters of high COVID-19 cases and deaths, and high calls to 211 for food assistance in the northwest areas of Detroit, where there are greater low-food-access pockets despite higher scoring NEMS stores than in other areas of the city [59,60]. On the other hand, the results also revealed that death rates are not predicted by case rates, food insecurity, or NEMS scores. This finding may need some more focused analysis within the specified ZIP code areas to understand why death rates were not predicted by these aligned variables. Though NEMS scores did not significantly contribute to predicting food insecurity, the relationship demonstrated was negative, indicating that there may be a protective influence of high-scoring stores. Further analyses should explore the potential effects of healthy grocery store environments on COVID-related outcomes.

This study provides a first assessment of the grocery food environment on COVID-19 cases, deaths and food insecurity among an urban population and shows potential for healthy grocery stores to serve as protective factors for incidence and mortality of COVID-19. Additionally, higher NEMS scores, or healthier grocery food environments, in Detroit were associated with fewer calls to 211 in the studied ZIP codes. This may mean that healthy grocers assisted communities with obtaining high-quality, healthy foods throughout the pandemic and can serve as a potential resource for food and nutrition security within these communities in the future. This is aligned with other research studies in the field, which have cited food stores as contributing to healthy diets within racial and ethnic minority communities and provides a premise for promoting healthy grocers in low-income communities of color as is occurring through the Great Grocer Project in Detroit [59,72,73]. 

Race is a significant influence on achieving full food security [15,16,74]. When controlling for demographic factors, including race, Burke et al. [74] found food insecurity persisted among racial and ethnic minority communities but not among people that were white, where power and privilege have historically resided [3]. This signifies that people who have been targeted by unjust policies or entrenched social norms within racial and ethnic minority communities consistently remain affected to a higher degree by unjust policies overtime when compared to their white peers, which can serve as a proxy for structural racism [3,74]. This aligns with the premise and findings of the Centers for Disease Control and Prevention’s (CDC) Social Vulnerability Index (SVI) which outlines the relationship between the Home Owners’ Loan Corporation policy influence from the 1930s—redlining—and the impact on present health outcomes and social mobility, as well as the readiness to respond and recover from public health emergencies, such as COVID-19 [75]. Given that the findings illustrated a relationship between food insecurity and COVID-19 cases and deaths, and inequities of these measures were more present among Black, low-income Detroiters, this highlights a “fundamental cause” of structural racism on food insecurity. It also provides a premise from which to continue to study food insecurity as a proxy for structural and systematic racism in nutrition research [76].

Policies and socially entrenched norms have prescribed the health-related quality of life outcomes for racial and ethnic minority groups, which was even more evident during the time of COVID-19 with more cases, hospitalizations and deaths inequitably impacting communities of color [3,5,77]. Historically, people from racial and ethnic minority groups have earned less in wages per hour and have held more ‘essential worker’ roles than their white peers [77]. The onset of COVID-19 was more problematic for racial and ethnic minority communities and for essential worker communities—a confounding dynamic—considering both communities are intertwined [3,77]. Further, prior to the pandemic, racial and ethnic minority communities and essential workers experienced food insecurity at higher rates than their peers [3,77]. The findings from this study outline the relationship between COVID-19 and food insecurity in the city of Detroit, which is home to primarily Black/African American, Hispanic/Latino and Arab immigrant families living under the constraints of structural racism, making them more susceptible to acquiring COVID-19. Future research could include the examination of the SVI index on these Detroit communities, as well as addressing the need for increasing access to high-quality, affordable healthy foods in grocery stores. Furthermore, future research should also assess how a healthier grocery environment can reduce and/or eliminate food insecurity and serve as a protective factor for structural racism within racial and ethnic minority communities [76]. 

There are a few limitations noted in this study. For example, there was a small sample size when conducting the regression analyses. However, since the regression was addressing a specific geographical area, the small sample size is less of a focus [78]. The study’s purpose was to analyze disparities in this geographical area and guide future studies to examine the impact of COVID-19 and food security in communities. Additionally, NEMS scores were averaged within each available ZIP code (N = 29, missing), which may have diminished important outcomes. Additionally, ZIP-code-level data comprise the single largest limitation of spatial analyses. City and county governments are wary of releasing health data at lower geographic levels, making ZIP code the de facto health geography unit when it is designed for postal delivery. Identifying clusters using ZIP codes is a useful descriptive test of autocorrelated phenomena when more granular data are not available. Future studies should examine individual scores of grocery stores and the impact high-scoring stores have on food security. Finally, causality cannot be determined from the relationships demonstrated in this study and further analyses over time would need to be conducted to determine a causal relationship.

## 6. Conclusions

This first assessment of grocery food environments and the relationship between COVID-19 cases and deaths and food insecurity as a proxy for structural racism within an urban setting provides a meaningful contribution to the literature. COVID-19 cases and deaths in Detroit were found to be associated with greater food insecurity among families. Higher NEMS scores, and the presence of more affordable, high-quality, healthy foods in grocery stores, also showed promise in reducing the impact of food insecurity, and potentially structural racism, among Detroit families. Further investigation into the healthfulness of grocery retail environments and the potential protective factors it can elicit in historically underserved communities is needed to determine reversal of inequitable health outcomes and lessoned burden from structural and institutionalized racism in low-income, racial and ethnic minority communities.

## Figures and Tables

**Figure 1 nutrients-14-02130-f001:**
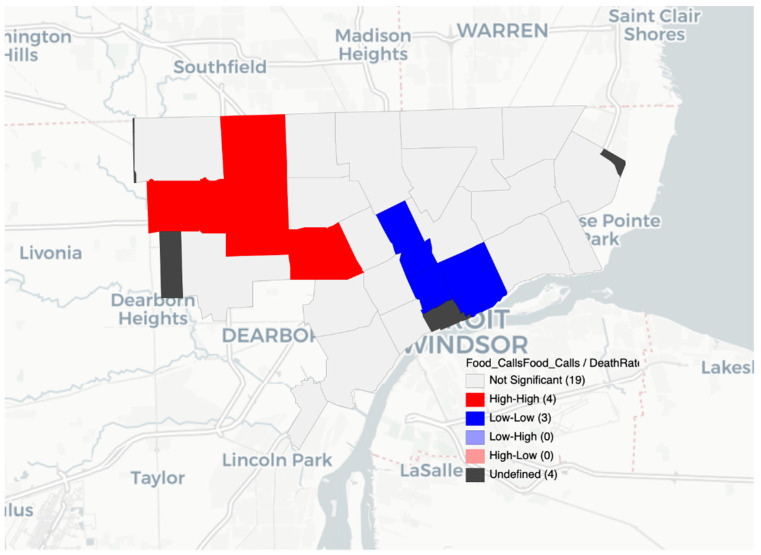
Food assistance calls from 211 and COVID-19 death rates by ZIP code in Detroit.

**Figure 2 nutrients-14-02130-f002:**
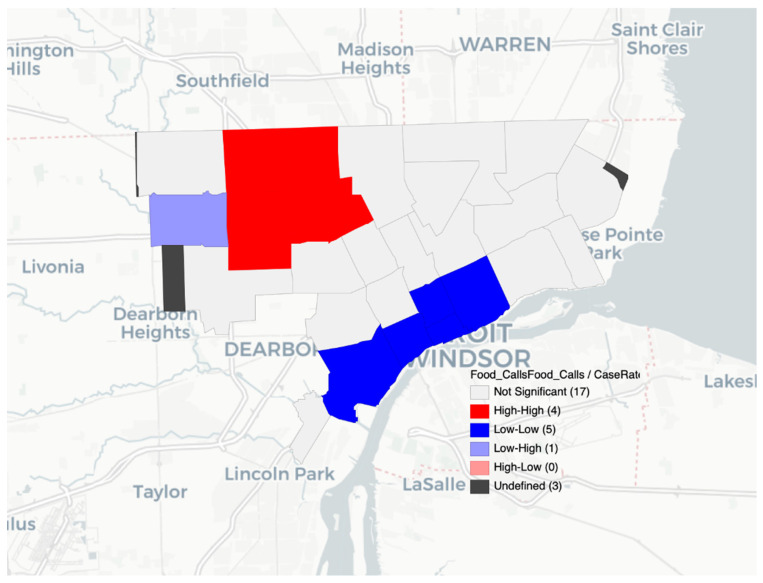
Food assistance calls from 211 and COVID-19 case rates by ZIP code in Detroit.

**Figure 3 nutrients-14-02130-f003:**
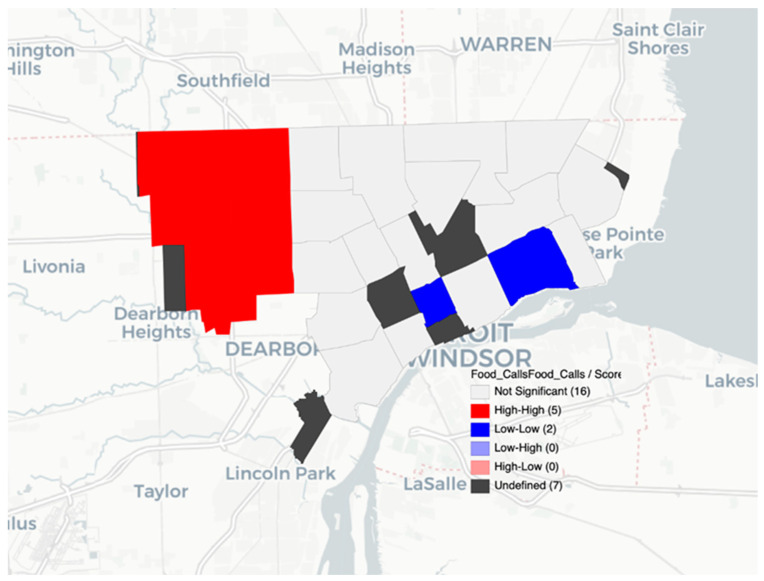
Food assistance calls from 211 and NEMS mean scoring by ZIP code in Detroit.

**Table 1 nutrients-14-02130-t001:** Results of the NEMS scoring: Means by ZIP code.

ZIP Code	Race (%Black/African American; %White) [67]	Quality (Range 0–6)	Availability (Range 0–27)	Price (Range −8–16)	Total (Range −8–49)
48201	70%; 20%	6.00	22.00	0.00	28.00
48202	82%; 12%	6.00	23.00	−2.00	27.00
48203	92%; 5%	6.00	27.00	−3.00	30.00
48204	97%; 1%	6.00	23.00	−2.00	27.00
48205	92%; 5%	5.00	19.33	1.00	25.33
48206	95%; 2%	6.00	19.00	4.50	29.50
48207	89%; 7%	5.14	22.29	3.00	30.00
48209	10%; 50%	5.25	17.00	3.25	25.50
48210	28%; 42%	6.00	16.50	3.50	26.00
48212	37%; 37%	6.00	18.00	1.00	25.00
48213	96%; 2%	6.00	25.00	3.00	34.00
48214	91%; 6%	6.00	26.00	2.50	34.50
48215	92%; 5%	4.50	22.00	2.00	28.50
48216	42%; 38%	6.00	18.00	−3.00	21.00
48219	91%; 7%	3.75	22.25	0.50	26.50
48221	93%; 4%	5.00	23.00	0.67	28.67
48223	89%; 8%	6.00	0.00	0.00	6.00
48224	90%; 8%	6.00	17.50	3.00	26.50
48227	96%; 2%	4.50	20.17	2.00	26.67
48228	79%; 17%	6.00	19.20	2.75	27.00
48234	94%; 4%	6.00	17.33	0.00	23.30
48235	97%; 1%	4.50	9.50	3.00	17.00
48238	97%; 1%	4.00	10.00	0.33	14.33

**Table 2 nutrients-14-02130-t002:** Results of regression analyses.

Regression	B	95% CI	*β*	*p*
Analysis 1				
Constant	136.97	−178.49, 452.40	-	0.37
Case Counts	0.047	0.008, 0.086	0.47	0.02
Death Counts	1.89	0.34, 3.43	0.47	0.02
NEMS	−0.36	−10.77, 10.06	−0.008	0.94
Analysis 2				
Constant	218.74	−159.85, 597.33	-	0.24
Case Rates	−0.002	−0.02, 0.02	−0.06	0.83
NEMS	3.05	−5.35, 11.44	0.17	0.45
Calls to 2-1-1	0.19	−0.03, 0.40	0.45	0.08

## Data Availability

Data reported in this study can be publicly found on the City of Detroit COVID-19 Dashboard at: https://detroitmi.gov/departments/detroit-health-department/programs-and-services/communicable-disease/coronavirus-COVID-19 (accessed on 28 February 2022). Calls to 211 can also be found on the United Way of Southeastern Michigan website at: https://app.powerbi.com/view?r=eyJrIjoiODRiYjJiZDAtZGUwMS00YTY1LTg1MWUtNWNhNzEwZTkzOGM0IiwidCI6ImIxNTJkZTI1LTYxZDMtNDlhMi1hMmY4LTczMWQ2ZTgxNDAyOSIsImMiOjN9&pageName=ReportSectiona99dfe17b930ed9503d2 (accessed on 27 February 2022). NEMS raw scores of stores are not shared publicly to protect the independent grocers in Detroit. Collective ratings of each store with a listing of each category met in the Great Grocer Project can be found on the Detroit Food Policy Council/Detroit Grocery Coalition website: https://www.detroitfoodpc.org/committees/#dgc (accessed on 6 March 2022).

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
