# Peer review of "Assessing the Influence of Food Insecurity and Retail Environments as a Proxy for Structural Racism on the COVID-19 Pandemic in an Urban Setting"

_nutrients, 2022, doi:10.3390/nu14102130_

Round 1
Reviewer 1 Report
The main problems of the manuscript are listed below:
The main aim of the study was to link structural racism with community Food Environments in the COVID-19 Pandemic.
While in the introduction (which was quite long) the authors talk about obesity, food insecurity, and racism, it seems to me that those three main bullet points were never associated with each other, but rather independently mentioned in the literature.
Lines 91-92 provide outdated information since we are in 2022 and the prediction was for 2021.
Methods
Quasi-experimental design lack randomness and usually uses either a comparison group or propensity score to match treatment and comparison groups. I am not sure if the MEMS scores were used for matching or what?
Sample: line 161 refers to the number of calls to 2-1-1 for food assistance since March of 2021. When were the beginning and the end of the data collection? It is not clear. Also, the database of COVI-19 was used, which period did the researchers use for the analysis? The same as for the # of 2-1-1 calls? It is not clear.
The variables (both dependent and independent) were not defined properly in the Methods section. The number of calls to 2-1-1 had increased during COVID-19 due to several factors: people lost their jobs, supermarkets experienced low inventory, children were not eating food provided by schools because they were at home, etc.
Variables ZIP codes: I believe the number of cases and deaths were used in the analyses based on the ZIP codes, however, no other explanation was given regarding socioeconomic status. Were those ZIP codes included in the analyses had predominantly African -Americans, Latinos/Hispanics, and immigrant families (lines 148-150)? If so, you need to include the ZIP Codes with Whites for a comparison group, or a dummy variable for gender.
GGP: in the analysis, the authors refer to GGP but haven’t provided any explanation on how GGP was developed. Is it the same as the NEMS scores? It is hard to identify.
In lines 162-165, the authors refer to the City-level data about COVID-19, but they never clearly specified what COVID-19 variable they were using? It was rates or the number of cases?
Analysis.
I am not convinced that regression analyses can be performed because of the quasi-experimental design. Authors reported no classical assumptions violations (lines 215-224), which are verified in classical OLS regression, where a sample is randomly selected, differently than in quasi-experimental designs.
Results
Overall, the presentation of results is very confusing and not clearly presented.
Table 1 – What is the relevance of table 1? How Quality, Availability, and Price ranges are ranked? A brief explanation of how those scores were calculated should be written in the section Materials and Methods. What does it mean a negative price score?
3.1.1 Regression 1 – Predicting Food Insecurity
Assuming the statistical analysis is accurate (which I doubt), lines 217-223 are irrelevant to the results because all classical assumptions were met. The model must be written as: Y = b0 + b1X1 = b2X2 + ... +bnXn, where Y is the dependent variable and Xn is the nth independent variable.
The results must be presented in a tabular form, including sample size, variables, parameter estimates, and p-values. Adjusted R2 is a relevant statistic, while F-test is not.
Model 1 : Number of calls = NEMS + Case Counts + Deaths Counts + GGP Store scores.
Covid-19 cases and deaths must be highly correlated, and I am very curious of what is the correlation coefficient between these two independent variables.
I am not sure what is the relevance of the ZIP Codes variable in the model. In lines 230-232: “Based on the ZIP CODE, as case counts of COVID-19 increased, so do calls for food assistance indicating higher levels of food insecurity”. Question: What actually do the zip codes measure? Is the distribution of the MEMS scoring presented in table 1 OR it is the number of COVID-19 cases and deaths by zip codes, as stated in line 164? Also, in line 181, it says: “… and COVID-19 cases, deaths and average GGP store scores (average store score per ZIP code) were compared…”
What is also confusing is that in line 226, it says, “the overall model was significant; case count, the death count and NEMS scores significantly ….” , but in 181 the authors refer to GGP store scores.
Lines 232 – 234 are irrelevant to the results. If an independent variable is NOT statistically significant in the model, that is H0: Bj = 0, then there is NO relationship between Y and X!!!!
At this point, I have stopped reading the manuscript because it has several inconsistencies and the statistical analysis of Model 1 is weak.
My main criticism of this paper is that the main objective of the study: structural racism with community Food Environments in the COVID-19 cases and deaths was NOT addressed in the statistical analyzes. The author did not show or prove ANY link between racism, food insecurity, and COVID-19 cases or deaths.
Author Response
Thank you for reviewing our manuscript. Our specific responses to your comments are below.
Reviewer 1
The main problems of the manuscript are listed below:
The main aim of the study was to link structural racism with community Food Environments in the COVID-19 Pandemic.
Reviewer Comment 1: While in the introduction (which was quite long) the authors talk about obesity, food insecurity, and racism, it seems to me that those three main bullet points were never associated with each other, but rather independently mentioned in the literature.
Response 1: Introduction was more cohesively connected and shortened (Lines 37-172)
Reviewer Comment 2: Lines 91-92 provide outdated information since we are in 2022 and the prediction was for 2021.
Response 2: Removed prediction sentence and restated food insecurity rate (Lines 92-94).
Reviewer Comment 3: Quasi-experimental design lack randomness and usually uses either a comparison group or propensity score to match treatment and comparison groups. I am not sure if the MEMS scores were used for matching or what?
Response 3: Design is a cross sectional design and not quasi-experimental. Study is assessing data at one point in time and not across two time points (Line 174).
Reviewer Comment 4: Sample: line 161 refers to the number of calls to 2-1-1 for food assistance since March of 2021. When were the beginning and the end of the data collection? It is not clear. Also, the database of COVI-19 was used, which period did the researchers use for the analysis? The same as for the # of 2-1-1 calls? It is not clear.
Response 4: Dates in which data was acquired from both databases were noted in the manuscript (Lines 192-194).
Reviewer Comment 5: The variables (both dependent and independent) were not defined properly in the Methods section. The number of calls to 2-1-1 had increased during COVID-19 due to several factors: people lost their jobs, supermarkets experienced low inventory, children were not eating food provided by schools because they were at home, etc.
Response 5: We reviewed 2-1-1 calls specifically for food security related information and we wanted to determine impact on these calls due to neighborhood grocery food environments (NEMS scores) and Covid-19 rates and deaths. This is why 2-1-1 was used as the dependent variable and Covid-19 cases and deaths as well as NEMS scores were the independent variable in the regression analyses. We further clarified and described the variables within the Methods section (Lines 175-203).
Reviewer Comment 6: Variables ZIP codes: I believe the number of cases and deaths were used in the analyses based on the ZIP codes, however, no other explanation was given regarding socioeconomic status. Were those ZIP codes included in the analyses had predominantly African -Americans, Latinos/Hispanics, and immigrant families (lines 148-150)? If so, you need to include the ZIP Codes with Whites for a comparison group, or a dummy variable for gender.
Response 6: We included demographic characteristics of the ZIP Codes in Table 1 (Lines 247-248).
Reviewer Comment 7: GGP: in the analysis, the authors refer to GGP but haven’t provided any explanation on how GGP was developed. Is it the same as the NEMS scores? It is hard to identify.
Response 7: GGP or the Great Grocer Project is the name of the intervention taking place within Detroit to promote healthy grocers. This was more thoroughly explained in the introduction (Lines 145-158). We also removed the reference to GGP in the methods and results to clarify that NEMS scores were used in the analysis.
Reviewer Comment 8: In lines 162-165, the authors refer to the City-level data about COVID-19, but they never clearly specified what COVID-19 variable they were using? It was rates or the number of cases?
Response 8: We clarified the description of the data used as both cases and rates were used in the two different regressions. Cases were used in Regression 1 and Rates were used in Regression 2 (Lines 190-194).
Reviewer Comment 9: I am not convinced that regression analyses can be performed because of the quasi-experimental design. Authors reported no classical assumptions violations (lines 215-224), which are verified in classical OLS regression, where a sample is randomly selected, differently than in quasi-experimental designs.
Response 9: We addressed this by clarifying that we are conducting a cross-sectional study vs. quasi experimental design (Line 174).
Reviewer Comment 10: Overall, the presentation of results is very confusing and not clearly presented.
Response 10: We clarified results presentation and included additional tables as well as more information in Table 1 (Lines 204-314).
Reviewer Comment 11: Table 1 – What is the relevance of table 1? How Quality, Availability, and Price ranges are ranked? A brief explanation of how those scores were calculated should be written in the section Materials and Methods. What does it mean a negative price score?
Response 11: This was addressed in the methods (Lines 176-189).
Reviewer Comment 12: 3.1.1 Regression 1 – Predicting Food Insecurity
Assuming the statistical analysis is accurate (which I doubt), lines 217-223 are irrelevant to the results because all classical assumptions were met. The model must be written as: Y = b0 + b1X1 = b2X2 + ... +bnXn, where Y is the dependent variable and Xn is the nth independent variable.
Response 12: We have usually reported the noted assumptions along with reporting results in manuscripts. The equation of the model is correctly written within the paper (Lines 270-271).
Reviewer Comment 13 Regression 1: The results must be presented in a tabular form, including sample size, variables, parameter estimates, and p-values. Adjusted R2 is a relevant statistic, while F-test is not.
Response 13: Table 2 was added to the paper (Lines 277-279).
Reviewer Comment 14: Model 1 : Number of calls = NEMS + Case Counts + Deaths Counts + GGP Store scores.
Response 14: NEMS and GGP Store scores are same variable. GGP or the Great Grocer Project is the name of the intervention taking place within Detroit to promote healthy grocers. This was more thoroughly explained in the introduction (Lines 145-158). We also removed the reference to GGP in the methods and results to clarify that NEMS scores were used in the analysis.
Reviewer Comment 15: Covid-19 cases and deaths must be highly correlated, and I am very curious of what is the correlation coefficient between these two independent variables. Kat will address.
Response 15: Correlations were included in Paragraph 1 of the results (Lines 240-246).
Reviewer Comment 16: I am not sure what is the relevance of the ZIP Codes variable in the model. In lines 230-232: “Based on the ZIP CODE, as case counts of COVID-19 increased, so do calls for food assistance indicating higher levels of food insecurity”. Question: What actually do the zip codes measure? Is the distribution of the MEMS scoring presented in table 1 OR it is the number of COVID-19 cases and deaths by zip codes, as stated in line 164? Also, in line 181, it says: “… and COVID-19 cases, deaths and average GGP store scores (average store score per ZIP code) were compared…”
Response 16: ZIP code usage as a variable in this study was clarified in the methods section (Lines 196-202).
Reviewer Comment 17: What is also confusing is that in line 226, it says, “the overall model was significant; case count, the death count and NEMS scores significantly ….” , but in 181 the authors refer to GGP store scores.
Response 17: GGP or the Great Grocer Project is the name of the intervention taking place within Detroit to promote healthy grocers. This was more thoroughly explained in the introduction (Lines 145-158). We also removed the reference to GGP in the methods and results to clarify that NEMS scores were used in the analysis.
Reviewer Comment 18: Lines 232 – 234 are irrelevant to the results. If an independent variable is NOT statistically significant in the model, that is H0: Bj = 0, then there is NO relationship between Y and X!!!!
Response 18: Several researchers have noted that p-values are arbitrary and results should not be ignored based on p-value alone. See Wasserstein & Lazar, 2016, Greenland et al., 2016 (links below). Further, it is imperative to include values which are not statistically significant to provide the reader with a “full picture” and to further the field of study.
https://amstat.tandfonline.com/doi/full/10.1080/00031305.2016.1154108#.Yk44AHrMJPY
https://pubmed.ncbi.nlm.nih.gov/27209009/
https://www.ncbi.nlm.nih.gov/pmc/articles/PMC6542161/
Reviewer Comment 19: At this point, I have stopped reading the manuscript because it has several inconsistencies and the statistical analysis of Model 1 is weak.
My main criticism of this paper is that the main objective of the study: structural racism with community Food Environments in the COVID-19 cases and deaths was NOT addressed in the statistical analyzes. The author did not show or prove ANY link between racism, food insecurity, and COVID-19 cases or deaths.
Response 19: In this study we use food insecurity and healthy food environments (grocery environments) as proxy measures for structural racism on COVID-19 cases and deaths in one urban environment. We note that more studies need to be conducted using these measures in other localities to determine causal links, which our study does not do. We highlighted this relationship and clarified our usage of food insecurity as a proxy measure in the introduction (Lines 106-109 and 159-172) and discussion (353-357).
Reviewer 2 Report
The authors have written an interesting paper on an important subject. It is a good fit for this journal. However, I suggest a few revisions.
First, the introduction could more clearly articulate the purpose of the paper in the context of the existing literature, research gaps, and how/why this method is instructive to fill these research gaps.
Second, the methods need to be more clearly explained: Why were these methods conducted? A better meta-explanation is needed: What are the insights and blind-spots that this suite of methods can (or cannot) provide? Additionally, the nitty-gritty is also missing here. I suggest that the authors expand the methods section to give the reader more detail, with a possible flow diagram to explain the logic of the methods. I think the methods are solid, but just need to be explained in greater detail to the reader, rather than being taken for granted that their logics make sense.
Third, the analysis of the data in the Results and Discussion sections could be expanded. What is the meaning of the statistical and geographical analyses? The authors could detail what they show and why they are important analytically from an empirical, theoretical, and methodological standpoint. In my opinion, the results do not speak for themselves, but rather must be explained so the reader can understand their value. This is especially the case with statistical and spatial analysis as the method is just the starting point, as the explanation and connection to theory or existing literature is key.
Fourth, in the Discussion or Conclusion section, more could be said about the connection between existing research and the research findings, especially for research beyond this specific case study. What can we understand that is new and important from Detroit that would inform new conceptual or methodological insights for existing research, future research, or policy frontiers?
Good luck on the revisions.
Author Response
Thank you for reviewing our paper. Specific responses to your comments are below.
Reviewer 2
The authors have written an interesting paper on an important subject. It is a good fit for this journal. However, I suggest a few revisions.
Reviewer Comment 1: First, the introduction could more clearly articulate the purpose of the paper in the context of the existing literature, research gaps, and how/why this method is instructive to fill these research gaps.
Response 1: The introduction was more clearly articulated and connected the components of food insecurity and grocery environments as a proxy measure for structural racism (Lines 106-109 and 159-172)
Reviewer Comment 2: Second, the methods need to be more clearly explained: Why were these methods conducted? A better meta-explanation is needed: What are the insights and blind-spots that this suite of methods can (or cannot) provide? Additionally, the nitty-gritty is also missing here. I suggest that the authors expand the methods section to give the reader more detail, with a possible flow diagram to explain the logic of the methods. I think the methods are solid, but just need to be explained in greater detail to the reader, rather than being taken for granted that their logics make sense.
Response 2: The methods for each type of analyses and measures used for each were clarified and explained more thoroughly throughout this section (Lines 174-203). Design was also clarified (Line 174).
Reviewer Comment 3: Third, the analysis of the data in the Results and Discussion sections could be expanded. What is the meaning of the statistical and geographical analyses? The authors could detail what they show and why they are important analytically from an empirical, theoretical, and methodological standpoint. In my opinion, the results do not speak for themselves, but rather must be explained so the reader can understand their value. This is especially the case with statistical and spatial analysis as the method is just the starting point, as the explanation and connection to theory or existing literature is key.
Response 3: A more thorough explanation of the results was added to the discussion (Lines 318-328).
Reviewer Comment 4: Fourth, in the Discussion or Conclusion section, more could be said about the connection between existing research and the research findings, especially for research beyond this specific case study. What can we understand that is new and important from Detroit that would inform new conceptual or methodological insights for existing research, future research, or policy frontiers?
Response 4: Connection to existing literature related to food insecurity as a proxy measure for structural racism was noted in the discussion (Lines 353-357). Future research was discussed more thoroughly in lines 372-375 and also 326-329.
Reviewer 3 Report
The work is interesting and documents a beautiful and commendable initiative, such as the GGP, in particular for large multi-ethnic cities which highlights how the covid has accentuated the discrimination already present for which it is necessary to remedy.
It is suggested to insert limitations in the abstract, already present in the discussion part and also to insert QR's in the methods or in the final part of the introductions, to be answered in the discussion part.
Author Response
Thank you for reviewing our paper. Specific responses to your comments are below.
Reviewer 3
The work is interesting and documents a beautiful and commendable initiative, such as the GGP, in particular for large multi-ethnic cities which highlights how the covid has accentuated the discrimination already present for which it is necessary to remedy.
Reviewer 3 Comment: It is suggested to insert limitations in the abstract, already present in the discussion part and also to insert QR's in the methods or in the final part of the introductions, to be answered in the discussion part.
Response: Limitations were inserted into the abstract (lines 23-24). The study purpose was clarified in the Introduction (lines 159-172) and addressed again in the Discussion (Lines 316-390) as we were assuming “QR’s” meant Research Questions and noted the research questions we were seeking to ask through this study in the sections mentioned above.
Round 2
Reviewer 1 Report
Even though the authors made several changes to the manuscript, in my scientific opinion, the manuscript does not bring enough evidence to support its title.
While the authors found a relationship between the number of 2-1-1 calls and Covid-19 cases and deaths, a comparison group must be included in the analysis.
What would be the results using ZIP codes with predominantly White residents? There are only three zip codes, with wites being the majority. There is a selection bias when ONLY including ZIP codes with mostly BLACK residents.
Table 2 has B (parameter estimate) and Beta. What is Beta in Table 2, and why is it relevant?
Authors keep insisting on reporting the non-violations of Classical Assumptions. Durbin-Watson test is used for autocorrelation, i.e., errors being correlated to each other. However, autocorrelation is an issue in time-series data. When reporting that the errors are independent (line 247), I ask myself: independent of what? Themselves or independent of independent variables (no endogeneity).
Lines 191-192, "Additionally, COVID-19 cases and death counts and rates
were included to examine the health disparities associated with structural racism among the identified ZIP codes"
Lines 237-238, . Further, COVID-19 death and case counts were significantly correlated (R = .78, p < .001), but death and case rates were not (R = .11, p = .32).
What does that mean? Did you include both in the analysis? Case count and case rates?
Lines 261-262 The constant was not statistically significant (p=.37) (Table 2).
The constant (intercept) is never vital in any regression analysis unless it measures the fixed cost.
I believe authors should change the title of the manuscript to be consistent with their findings.
Author Response
Thank you for Reviewing our manuscript for a second round. Please see our responses below.
Comment: Even though the authors made several changes to the manuscript, in my scientific opinion, the manuscript does not bring enough evidence to support its title.
Response: Title was revised to align with primary study outcomes.
Comment: While the authors found a relationship between the number of 2-1-1 calls and Covid-19 cases and deaths, a comparison group must be included in the analysis.
What would be the results using ZIP codes with predominantly White residents? There are only three zip codes, with wites being the majority. There is a selection bias when ONLY including ZIP codes with mostly BLACK residents.
Response: We assessed all ZIP codes within the City of Detroit. Since the demographics of Detroit are 89% African American/Black there are no predominantly White ZIP codes in the city to compare to. Our study is assessing the impact of the Detroit grocery food environments and food insecurity on COVID-19 cases and rates. Assessing White ZIP codes will not be possible within this study as they do not exist in the City of Detroit.
Comment: Table 2 has B (parameter estimate) and Beta. What is Beta in Table 2, and why is it relevant?
Response: We have included the standardized coefficient in Table 2 as it is easier for some readers to interpret.
Comment: Authors keep insisting on reporting the non-violations of Classical Assumptions. Durbin-Watson test is used for autocorrelation, i.e., errors being correlated to each other. However, autocorrelation is an issue in time-series data. When reporting that the errors are independent (line 247), I ask myself: independent of what? Themselves or independent of independent variables (no endogeneity).
Response: We added a clarifying statement to indicate “of observations” into line 248.
Comment: Lines 191-192, "Additionally, COVID-19 cases and death counts and rates
were included to examine the health disparities associated with structural racism among the identified ZIP codes"
Lines 237-238, . Further, COVID-19 death and case counts were significantly correlated (R = .78, p < .001), but death and case rates were not (R = .11, p = .32).
What does that mean? Did you include both in the analysis? Case count and case rates?
Response: Both cases and rates were included in the analysis. This was indicated in the methods section in the first revision as well as within each reported table and figure in the results section.
Comment: Lines 261-262 The constant was not statistically significant (p=.37) (Table 2).
The constant (intercept) is never vital in any regression analysis unless it measures the fixed cost.
Response: Sentence was removed (line 262).
Comment: I believe authors should change the title of the manuscript to be consistent with their findings.
Response: Title was revised.
This manuscript is a resubmission of an earlier submission. The following is a list of the peer review reports and author responses from that submission.